# Niche Dynamics Below the Species Level: Evidence from Evaluating Niche Shifts within *Quercus aquifolioides*

Li Feng , Lipan Zhou, Tianyi Zhang and Xumei Wang *

School of Pharmacy, Xi'an Jiaotong University, Xi'an 710061, China
* Correspondence: wangxumei@mail.xjtu.edu.cn

**Abstract:** The role of ecological niches in lineage diversification has been the subject of long-standing interest of ecologists and evolutionary biologists. Specific responses to climate change can arise below the species level, resulting in differentiated adaptation or movement patterns within a given species. Thus, the urgent need to model potential responses to ongoing climate changes among genetically distinct populations within a species is increasingly recognized. In this study, we utilize the information of intraspecific variation within *Quercus aquifolioides* as a priori, and then focus on the potential distribution ranges and niche dynamics of its three intraspecific lineages (WSP, HDM, and Tibet) under current environmental conditions via ecological niche models (ENMs) and PCA-env ordination method, respectively. Our results indicated that the three lineages have occupied differentiated climatic niches. Although the three lineages have distinctly adaptive strategies for homogeneous environmental conditions, some lineages had sympatric projecting areas. The PCA-env demonstrated that the lineage pair WSP vs. HDM had the largest niche overlap while Tibet vs. HDM showed the smallest one. Moreover, the hypothesis of the niche was indistinguishable within the three lineages rejected, indicating the presence of niche divergence rather than niche conservatism below the species level. Our findings highlight the potential of modeling intraspecific responses to climate change and provide insights into lineage diversification within *Q. aquifolioides*, permitting the exploration of the information determined by niche evaluations and comparisons to understand plant diversification processes below the species level in biodiversity hotspots.

**Keywords:** diversification; ecological niche models; intraspecific level; niche evaluation; PCA-env; *Quercus aquifolioides*



## 1. Introduction

Ecological niche reflects not only local adaptation but also the tendency of related lineages and/or species to share environmental tolerances, which is important in understanding patterns of distribution and biodiversity, the discrimination of special habitats to buffer the impacts of global climatic change, and the reconstruction evolutionary trajectories of organisms [1]. However, due to the differences between the effects of genotype and phenotype within a species, the response of species to environmental change is not always identical across its distribution range [2]. Even though the majority of exercises focus on niche estimation treat species as a single unit, assuming that the data on species' current distributions is enough to reflect environmental tolerances, regardless of its response to climate change which will typically vary across space [2–5]. Therefore, knowledge of population structure and/or phylogenetic relationships must be integrated into ecological niche models (ENMs) and should be treated a priori when modeling species distributions, speculating niche dynamics, and forecasting the impacts of future climatic conditions on species distribution [1,6,7]. Not including this information may misestimate sub-taxon niche variation and have biased interpretations [6], especially for distinct lineages occupying discrete niche spaces [8].

Empirical studies have demonstrated that the rates of niche evolution vary tremendously below the species level [1,9]. Valladares et al. [2] have also indicated that the ecological niche models based on populations or lineages may be especially important, as they can identify spatial variation in genotypic or phenotypic traits below the species level. Accordingly, an increasing appreciation for the role of evolutionary processes, such as niche conservatism and local adaptation, has led to the modeling of niches below the species level [7,10–13]. These studies evaluated intraspecific niche dynamics and demonstrated the utility of modeling intraspecific responses to changing climates. For instance, integrating intraspecific variability significantly increases model sensitivity but decreases model specificity. To our knowledge, there is no similar study on plants inhabiting China, but one case on niche conservatism of *Gynandropaa* frogs [14].

Empirical studies have proposed that niches of descendant species were conservatism compared with their ancestors because descendant species tended to inhabit similar geographical areas and/or environmental conditions to their ancestors [15–17]. Quantitative approaches to niche evolution to indicate niche dynamics highly depend on the niche tests such as niche similarity and niche equivalency tests [9,18]. The niche equivalency test helps determine the transferability of niche models along a spatial-temporal dimension, while the niche similarity test is usually employed for testing biogeographic and evolutionary hypotheses [9]. Furthermore, Peterson [9] reviewed that recent and short-term events such as species invasions or distributional shifts at the end of the Pleistocene would lead the niche toward conservatism, while the longer-term events such as differentiation across phylogenies would show a tendency for a breakdown in niche conservatism. The niche conservatism here means the niches of descendant species tend to remain similar to their ancestor rather than remaining strictly equivalent [19]. However, the roles of niche conservatism versus niche divergence in promoting lineage diversification within species inhabiting diverse habitats remain elusive.

*Quercus aquifolioides* is an evergreen sclerophyllous oak species belonging to the *Quercus* section *Heterobalanus* [20]. Its range locates in the Sino-Himalayan Forest subkingdom with broad elevational gradients [21], e.g., across western Sichuan, northern Yunnan, and eastern Tibet (Figure 1), implying strong adaptation to different environments. A recent phylogeographical study has illuminated that three distinct lineages, i.e., lineages of Tibetan (hereafter called Tibet), West Sichuan Plateau (hereafter called WSP), and Hengduan Mts. (hereafter called HDM), exist within this species using cpDNA markers [22]. Numerous studies have demonstrated that the interspecific niches within oak species are convergent or divergent [23,24], while the intraspecific niche dynamics are yet poorly understood. Hence, *Q. aquifolioides* provides a suited model to investigate the niche dynamic below species.

In this study, we focus on the issue of whether the intraspecific lineages of *Q. aquifolioides* kept similar niches after their splitting, and explore whether the niches among these three lineages are conserved during their diversification process. Specifically, we constructed ecological niche models (ENMs) for each lineage, employing six methods (please see details below) implemented in the R package *biomod2* [25] based on the occurrence points in concert with pseudo absence data of each lineage. Then we investigated the niche overlap, niche similarity, and niche equivalence between these closely related lineages using the PCA-env approach developed by Broennimann et al. [18]. Finally, following the applied niche comparison framework by Guisan et al. [26], we distinguished the three basic components (e.g., niche unfilling, niche stability, and niche expansion) for paired comparison, accounting for the availability of environmental conditions within the respective ranges. Our findings will uncover a new and multifaceted view of climate niche shifts within *Q. aquifolioides*, and shed light on the role of intraspecific niche dynamics in producing lineage diversification in oak species.

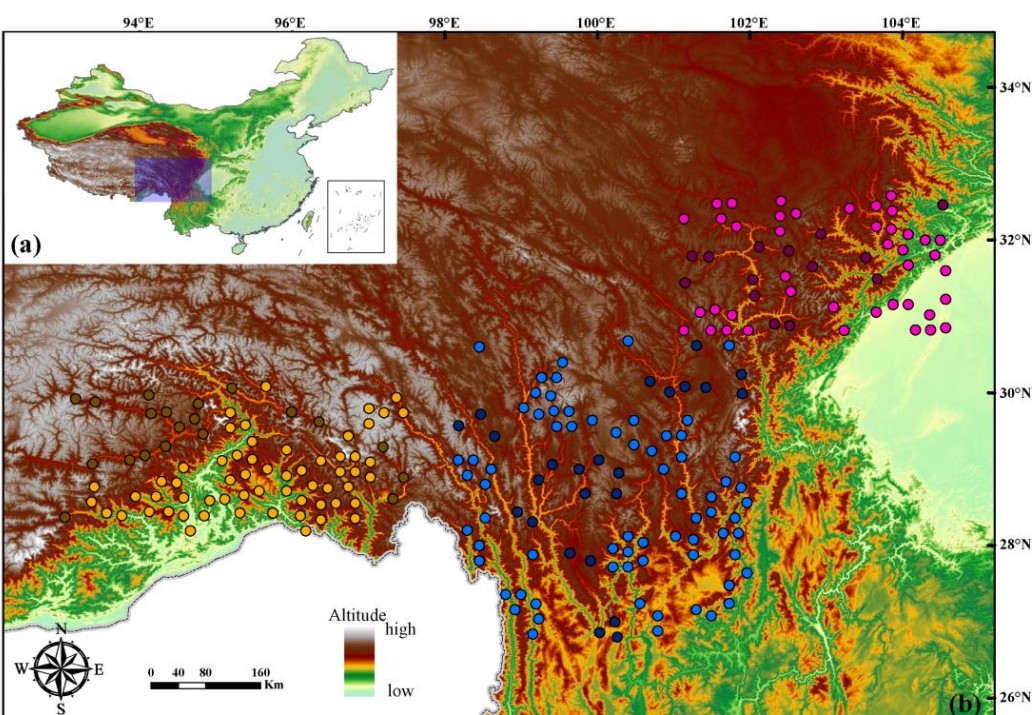

**Figure 1.** Distribution ranges of the three intraspecific lineages within *Quercus aquifolioides*. Points with dark yellow, blue, and pink represent the occurrences of lineages Tibet, WSP, and HDM, respectively. Points with yellow, blue, and pink represent the pseudo absences of lineages Tibet, WSP, and HDM, respectively.

## 2. Materials and Methods

### 2.1. Species Occurrences and Bioclimatic Data

Species occurrence data were gathered from Du et al. [22], and 58 populations across the distribution ranges of *Q. aquifolioides* were included in the present study (Table S1). We treated the clade identifications by four cpDNA loci (e.g., *trn*H-*psb*A, *rps*16, *trn*S (GCU)-*trn*T (GGU), and *trn*Q (UUG)-*trn*S (GGU)) in Du et al. [22] as a priori for representing the most appropriate phylogenetic level for ENMs (Figure S1). Consequently, the 58 occurrence data were then assigned to the three clades (e.g., the HDM, WSP, and Tibet lineages which involved 24, 14, and 20 points, respectively). Although a recent study has indicated that the absolute minimum sample sizes of 13 are enough for modeling distribution ranges of widespread species [27], given the absence data largely impacts the reality of ENMs results [28–30], we employed the R package *mopa* [31] to sample pseudo-absences points for each lineage from the accessible background (e.g., the range for HDM is from 93° E–97.5° E and 28.3° N–30.1° N, for WSP is from 101.1° E–104.6° E and 30.8° N–32.5° N, and for Tibet is from 98.1° E–102° E and 26.8° N–30.9° N, respectively), using the default settings except for modifying the parameters of *exclusion.buffer* (equal to 0.249) and *prevalence* (equal to −0.5). Our final dataset comprised 96 georeferenced occurrences for HDM lineage, 80 for Tibet lineage, and 56 for WSP lineage (Figure 1).

Empirical studies have illuminated that species richness and diversity patterns are severely affected by climate conditions where they occurred, especially temperature and precipitation variables [32–34]. In the present study, we used the 19 bioclimatic variables of the average of the period 1970–2000 developed by Fick and Hijmans [35] at 2.5-min spatial resolution (~5 km close to the equator), in order to derive a series of meaningful predictors that are crucial for plant growth and development. As strong co-linearity between environmental variables could inflate model accuracy in ENMs, we removed one of any two variables that was highly correlated (Pearson correlation > 0.8) on the basis of *ecospat* package [36], keeping the more biologically relevant variables (Figure S2). Hence, integrating the results of Pearson's correlation tests and a jackknife analysis for retaining

variables with higher permutation importance [37] (Table S2, Figure S3) to reduce the number of predictor variables included in the final models. We retained seven variables after the aforementioned steps: mean monthly temperature range (bio2), isothermality (bio3), temperature seasonality (bio4), temperature annual range (bio7), mean temperature of warmest quarter (bio10), annual precipitation (bio12), and precipitation of driest month (bio14).

### 2.2. Species Distribution Modeling

Based on the occurrences and pseudo absences as well as the seven selected variables, we modeled the current potential habitats for the *Q. aquifolioides* at lineage and species levels, respectively, using the R package *biomod2* [25] in R 3.4.8 [38]. The package *biomod2* offers multi-method modeling that generates the probability of presence outputs for each of the modeling approaches as well as a series of ensemble projections. Hence, this package attempts to combine the strengths of multiple modeling algorithms while accounting for their uncertainties [25].

Although this package includes ten available modeling algorithms, we selected six models (i.e., generalized linear model (GLM), generalized boosted regression model (GBM), surface range envelope (SRE), multivariate adaptive regression spline (MARS), random forest (RF), and maximum entropy (MAXENT)) in the present study given the computational speed, model prevalence and accuracy [39,40]. These models focus on the current potential habitats at and below the species levels for our focal species, calibrating on a random sample of the initial data and tested on the remaining datasets with both the receiver operating characteristic (ROC) curve and the true skill statistic (TSS) [41]. Given the ensemble results tend to generate more robust predictions and reduce model uncertainties [42], thus predicted probabilities from the individual models in each model cluster were integrated as a consensus model, combining the median probability over the selected models with TSS > 0.7 [25]. We chose the median value for the TSS because it is less sensitive to outliers when comparing with the mean value [43]. The entire procedure was repeated 120 times (20 times for each model) for each lineage to ensure the robustness of predicted distribution ranges and provide uncertainty estimates.

### 2.3. Bioclimatic Variable Analyses

Two methods were used to determine ecological differentiation among the three lineages. We extracted values for each of the seven bioclimatic variables for all 58 occurrences using the R package *raster*. First, the lineage-level divergence associated with each of the environmental variables was examined by a nonparametric Kruskal–Wallis multiple-range test [44]. The distribution of each lineage in line with specific environmental variables was displayed in violin plots based on the R packages *ggplot2* [45] and *ggpubr*. Then we conducted a principal component analysis (PCA) to compare niche differentiation between lineages to determine whether the three clades were ecologically differentiated.

### 2.4. Niche Equivalence and Niche Similarity Tests

We utilized the PCA-env method introduced by Broennimann et al. [18] to evaluate niche dynamics between the lineages of *Q. aquifolioides*. Firstly, this method translates a multivariable environmental space into a two-dimensional one using a PCA. Then the PCA space (100 × 100 grid-cell resolution) is gridded and smoothed densities of the data (i.e., occurrences of the three lineages), the smoothed densities of the study area (in our case, the combined data of occurrences, and pseudo-absences for the three lineages each), are also calculated under the same cell resolution using a kernel density function. Finally, the species' density function is divided by the study areas to derive a description of the realized climate niche accounting for environmental availability in the study area [26].

Niche overlap between lineages is measured in the light of Schoener's *D* [19,46], ranging from 0 (no overlap) to 1 (complete overlap). The observed niche overlap is then compared to random measures of niche overlap. We also applied the niche equivalency

and the niche similarity tests [47,48]. They are frequently used statistical tests considering hypotheses of niche divergence or conservatism. The niche equivalency test is conservative as it only compares the modeled niches of the two species and asks whether they are indistinguishable from each other [15,19]. The niche similarity test, however, calibrates niche models for one species (or lineage, in our case) and then predicts other species' occurrences better than expected by chance, accounting for the differences in the regions where both species occupied [49]. Although they have different null hypotheses [18,50], they are both tested based on comparisons of observed measures of niche overlap to random overlap values. Here, as we aimed to evaluate the niche dynamics within *Q. aquifolioides*, we set parameter *alternative* = "*greater*" or "*lower*" in the function of *ecospat.niche.equivalency.test* implemented in the R package *ecospat* to test if niches are more similar (similarity test) or different (equivalency test) than expected by chance. We used 1000 permutations for both tests to evaluate the significant levels ($\alpha = 5\%$). It is important to keep in mind that the niche equivalence and similarity tests only can investigate whether niche shifts have occurred rather than demonstrate their causal mechanisms.

We quantified the three niche dynamic metrics of niche unfilling, stability, and expansion between the three lineages via the R package *ecospat* to depict a holistic picture of the niche dynamic of these lineages beyond the overlapping regions. Previous studies had indicated that this analysis should be based on the shared analog climate between the two ranges being considered to gain meaningful interpretation for niche dynamics [26,51]. The niche expansion and stability always add up to 1, and niche conservatism is a tendency for species to maintain their niches in spatiotemporal scales, which is commonly used as niche stability [26].

## 3. Results

### 3.1. Lineage and Species Niche Models

ENMs for lineages and species had great predictive accuracies as measured by the values of ROC and TSS (ROC: 0.991–0.998; TSS: 0.956–0.993). Bioclimatic variable significance or contributions differed between the four datasets as well as among models within the four datasets (Table S3). Likewise, the average values of ROC and TSS among the six models that evaluated the discriminatory power of model predictions at present climatic data for the four datasets also differed between and/or within them (Figure 2). However, the RF and Maxent models seemed as optimal for all four datasets, given these two models both tended to be more predictive than the other four models and the reliable results corresponded with the collection records.

Although slightly over-predicted in portions near the Sichuan Basin for all three lineages, the ensemble ENMs for each of the three lineages matched their known distribution, respectively. The projected areas for lineage HDM were mainly located near the margin of southeast QTP (e.g., Hengduan Mts.), the potential areas for lineage Tibet were sparse across the QTP and Hengduan Mts., and the areas for lineage WSP situated near the west edge of Sichuan Basin (Figure 3). Among the three lineages, if we used the threshold above 0.5 to evaluate their areas, the projected area for lineage HDM (*c.* 124.1 thousand km$^2$) was much larger than the other two lineages. The lineage Tibet had moderate areas (*c.* 76.4 thousand km$^2$), while the lineage WPS had the smallest potential distribution (*c.* 63.7 thousand km$^2$). In addition, our ENMs' result indicated the projecting potential area for *Q. aquifolioides* was mainly located within the Hengduan Mts. and sparsely distributed within the QTP. The projected areas with moderately high suitability scores (>0.50) for *Q. aquifolioides* were *c.* 198.8 thousand km$^2$.

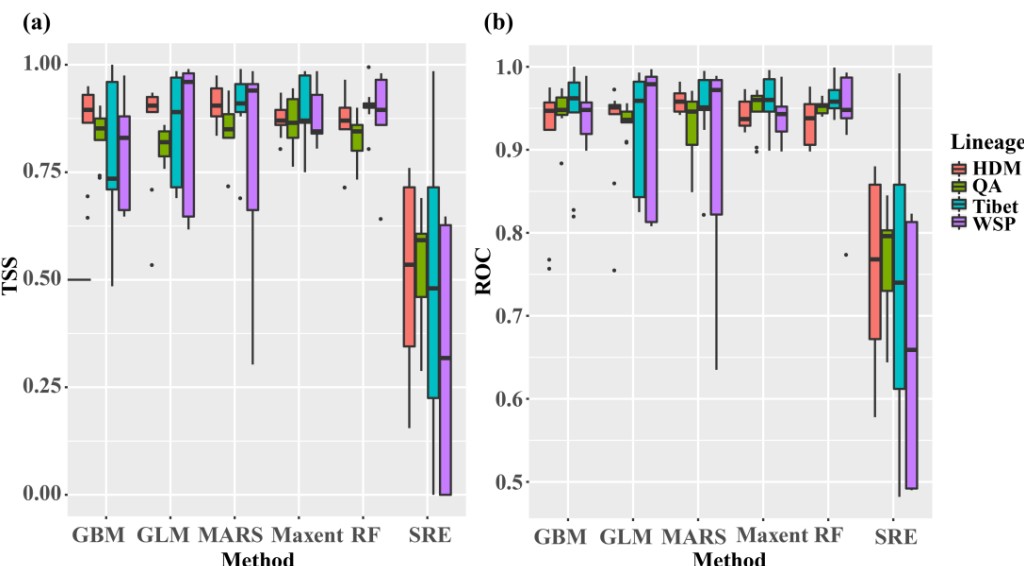

**Figure 2.** The results of TSS (**a**) and ROC (**b**) when evaluating single models for the distribution ranges of the three lineages of *Quercus aquifolioides* and this species with current climatic variables. HDM, Hengduan Mts.; QA, *Quercus aquifolioides*; WSP, West Sichuan Plateau.

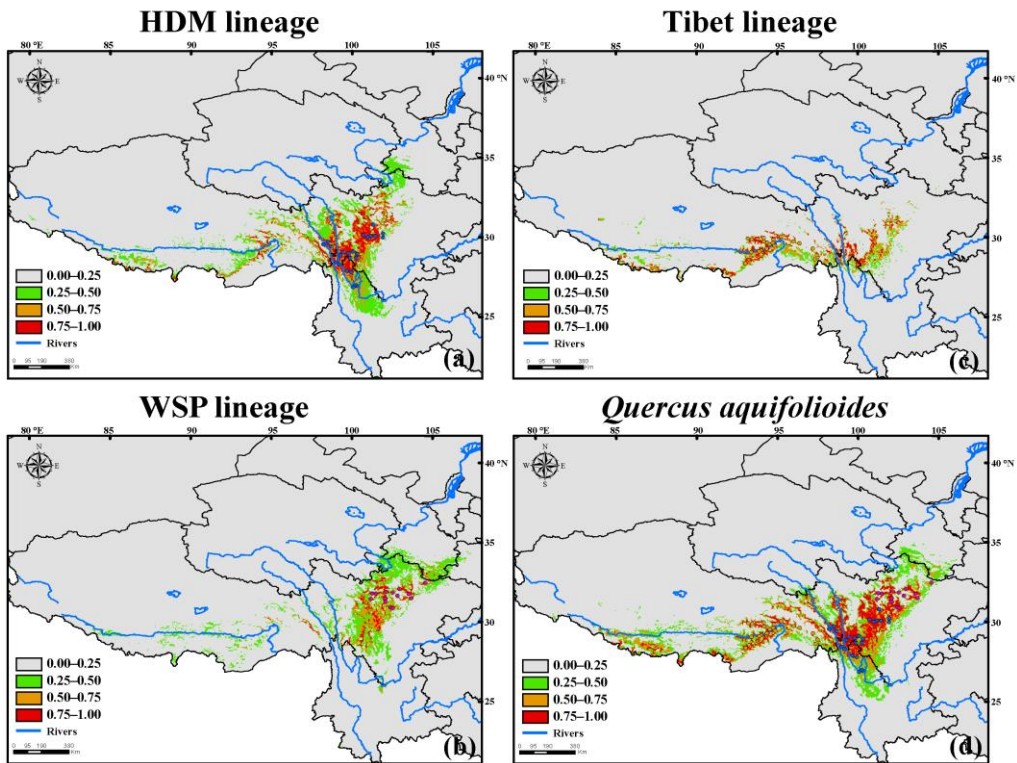

**Figure 3.** ENMs' results of (**a**–**c**) the three intraspecific lineages of *Quercus aquifolioides* and (**d**) this species at contemporary climatic conditions.

### 3.2. Bioclimatic Variable Analyses

Our nonparametric Kruskal–Wallis multiple-range tests revealed that the majority of the bioclimatic variables did not have significant differences among the three lineages, except when comparing differences between lineages Tibet and WSP based on the data extracted from the variables of bio2, bio4, and bio7, and between the lineages HDM and WSP at bio10 (Figure 4a–g). In addition, although the PCA showed that 99.87% of the total variance was explained by the first two principal components (PC1: 82.55% and PC2:

17.32%, respectively; Figure 4h), the PCA result indicated that the three lineages had lots of overlapping regions rather than distinguishable clusters each.

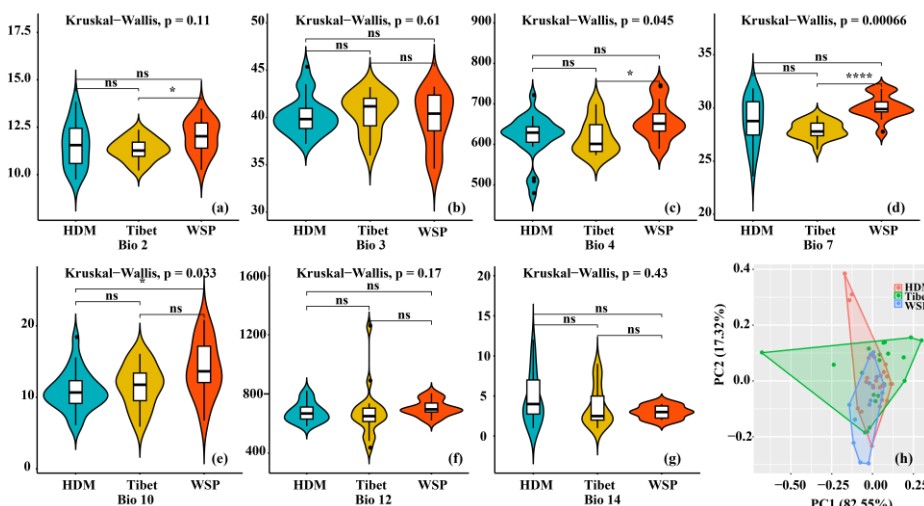

**Figure 4.** Nonparametric Kruskal–Wallis multiple-range tests (**a**–**g**) and the PCA (**h**) based on climatic data extracted from the 58 occurrences of three lineages belonging to *Quercus aquifolioides*.

### 3.3. Niche evaluation for the Three Lineages

Generally, our niche evaluation demonstrated that the three lineages occupied similar niche space, and Schoener's $D$ uncovered moderate levels of niche overlap between native and exotic ranges. Among the three lineage pairs, the niche overlapping of lineage pair WSP and HDM ($D = 0.638$) was highest followed by the pair WSP and Tibet ($D = 0.635$), and the pair Tibet and HDM had the smallest niche overlap value ($D = 0.498$). For all the pairs, the results of niche equivalence tests were all insignificant, therefore the hypothesis that any lineage pair is distributed in identical environmental space could not be rejected (Figure 5(a4,b4,c4)). However, in light of the results of niche similarity tests, the clade pairs HDM and WSP as well as WSP and Tibet yielded significant results, implying that the niches were less similar than random in two reciprocal directions, respectively. Interestingly, the niche similarity test showed that the niche had severe alternation in the direction of Tibet to HDM but not vice versa, leading to non-rejection of the null hypothesis of retained niche similarity for HDM to Tibet (Figure 5(a5,a6,b5,b6,c5,c6)).

Finally, the niche dynamics results showed that for all the lineage pair comparisons when considering two reciprocal directions, disregarding the values of niche unfilling, no one pair for the value of niche stability could be up to 1 (Table 1). Furthermore, the lineage WSP had the largest niche expansion and smallest niche stability in its exotic regions, while the lineage HDM had the smallest niche expansion and largest niche stability.

**Table 1.** Comparisons of the niche dynamics by pairs of the three lineages of *Quercus aquifolioides*, based on PCA-env analysis.

| Distribution Ranges (Comparisons A → B) | | Niche Dynamics Metrics | | |
|---|---|---|---|---|
| A | B | Expansion | Stability | Unfilling |
| WSP | Tibet | 0.28 | 0.72 | 0.14 |
|  | HDM | 0.24 | 0.76 | 0.07 |
| HDM | Tibet | 0.12 | 0.88 | 0.25 |
|  | WSP | 0.07 | 0.93 | 0.24 |
| Tibet | HDM | 0.25 | 0.75 | 0.12 |
|  | WSP | 0.14 | 0.86 | 0.28 |

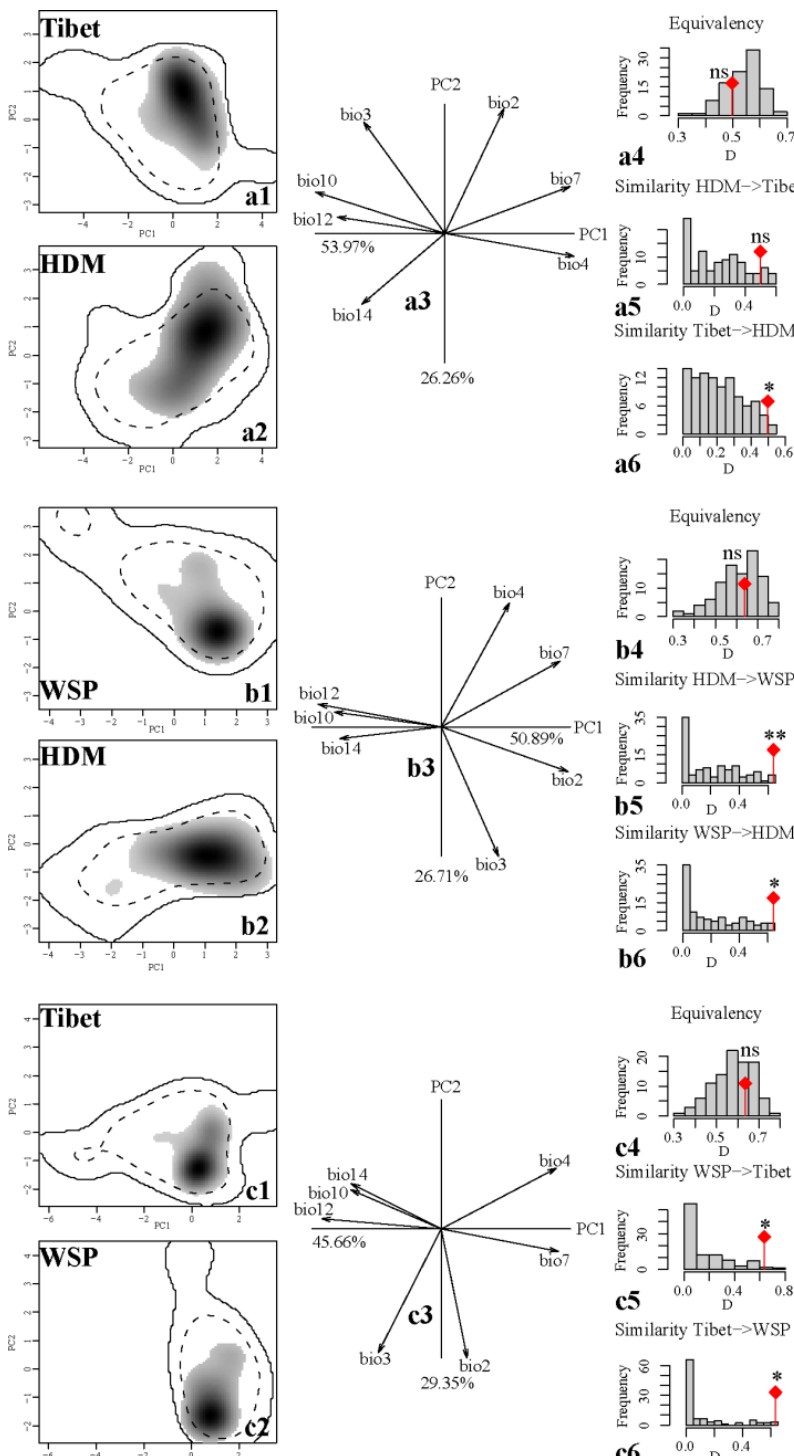

**Figure 5.** The niche dynamics of the three compared pairs within *Quercus aquifolioides* in climate space based on a principal component analysis (PCA-env). Panels (**a–c**) are the niche features of the three compared pairs along the two first axes of the PCA, respectively. In panels (**a1,a2,b1,b2,c1,c2**), shadow areas colored in gray represent the density of the occurrences of the lineage by cell. The 100% and 50% of the available environment are delimited by solid and dashed contour lines, respectively. Panels (**a3,b3,c3**) indicate the contribution of the environmental variables on the first two axes of the PCA and the percentage of inertia interpreted by the PCA axes. Histograms in a4-6, b4-6, and c4-6 represent the observed niche overlap (*D*) between the two lineages (bars with a diamond) and simulated niche overlaps (gray bars) based on the tests of niche equivalency and niche similarity, respectively. The significance of the tests is shown (ns, non-significant; * 0.01 < *p* < 0.05; ** *p* < 0.01).

## 4. Discussion

### 4.1. Comparisons of the ENM Distributions between Distinct Lineages and Species

Adaptation of populations to local circumstances appear to frequently occur, thus dividing species into subpopulations (e.g., defining by genetic variation based on the population-genetic methods) and modeling each as a distinct entity, which may reflect their specific evolutionary trajectories [1,8,52]. For instance, when modeling species potential distribution at sub-taxon levels, Valladares et al. [2] found some subpopulations had more restricted distributions compared with entire species while others were opposite, and they inferred the differed impacts of modeling fundamental versus realized niches which might lead to these differences.

In the present study, our results demonstrated that the realized niches which belong to the three intraspecific lineages were generally more limiting than that of the species (Figures 1 and 5), suggesting the limited lineage data may lead to restricted estimates of the peripheral climate niche. This may be also true when interpreting the differences in ENMs' results among the three distinct lineages. For instance, the split models indicated that the lineage HDM covered a much larger potential distribution than that of the other two lineages, and it also involved the most occurrences when modeling its distribution. When choosing 0.50 as the threshold suitability, the potential area of lineage HDM approximated the sums of distribution of lineages Tibet and WPS, while it was similar to the distribution ranges at the species level although some subtle differences existed. (Figure 5). Empirical studies have illuminated that the split models (lineage models in our case) generally estimate narrower distribution and/or niches than their lumped counterparts (species model in our case), but splitting does not necessarily result in smaller range estimates [1]. However, the effects of lineages correspond to generally well-defined, and often minimally overlapping climate niches could not be ruled out when explaining our ENMs' results, such as the case of Maguire et al. [7]. Additionally, currently projected distributions of the three lineages derived from the consensus model via *biomod2* only had moderate degrees of overlapping regions (Figure 3), indicating that each lineage occupies a unique geographic space and undergoes adaptive evolution. Likewise, the results of preferred bioclimatic variables when conducting split model (Table S3) for lineage HDM demonstrated that the top three climate variables in terms of variable importance identified both by MAXENT and RF models were bio4, bio10, and bio12, and the bio4 was the most crucial variable, while the bio7, bio12, and bio14 were preferred by Tibet lineage and the bio7 ranked first. For lineage WSP, two of the top three climate variables defined by the above two models were the same, including bio4 and bio14, the third variable was bio12 in the MAXENT model and bio7 in the RF model, in which bio12 ranked seventh, however, the most important climate variable identified by both models was the same (bio4). The aforementioned results also illuminated that the three intraspecific lineages evolved distinctly climate-adaptive trajectories in their native ranges.

### 4.2. Intraspecific Niche Dynamics Evaluations among the Three Lineages

The niche equivalency test did not reject the hypothesis that any lineage pair was distributed in equivalent environmental space, and the majority of niche similarity tests rejected the hypothesis that the observed niche overlap between the distinct lineages could be explained by the regional climate similarity of the two compared ranges (Figure 5). Although our simulated niche overlapped values, which were generated by random shifts of the entire shape of the clade's niche over the lineage's background area using the PCA-env, providing a simpler environmental space in which niche differences are obvious [18], it seemed plausible when given the three clades of our focal species occupied areas with severe changes in climate and topography inhabiting the southeastern QTP and adjacent Hengduan Mts. The niches for the three paired comparisons are not strictly equivalent (Figure 5, $p > 0.05$), which illuminated the three clades of *Q. aquifolioides* are likely to have adapted to local environments. In the viewpoint of niche conservatism, it could be specu-lated that the three intraspecific lineages did not significantly retain their environmental

niche characteristics from their ancient populations. In line with previous studies [14], our results indicated that the closely related clades did not have strictly equivalent niches under their local environments, and their niches typically were more dissimilar than expected given the environmental availabilities for them. In addition, a previous study has indicated that niche differentiation could arise due to the impact of secondary contact when competition occurs between different taxa with differentiated climate niches, which subsequently will exclude weak competitors in some regions or environments [53]. According to the results of Du et al. [22], this species has expanded its distribution ranges since the Last Interglacial, and the three lineages have numerous introgression determined by nuclear microsatellite (nSSR) markers, all of which may suggest that competitive exclusion may play an important role for the niche differentiation of the three intraspecific lineages.

Niche dynamic indices indicated that regardless of the debate of native and exotic lineages, all lineages had moderate degrees of niche expansion and considerable niche stabilities in exotic regions (Table 1), the niche stability was high overall (Stability $_{mean}$ = 0.82 ± 0.08) while the niche expansion was low (Expansion $_{mean}$ = 0.18 ± 0.08). Accordingly, the niches of the three intraspecific lineages were not strictly conservative. Niche conservatism may also be related to biotic traits, with functional traits of species with phylogenetic signals co-varying with climate niches. That is, lineages track their optimal niche space through time and move as environmental change arises [54]. *Quercus virginiana* and *Q. geminata* performed differently in terms of photosynthetic performance and growth, corresponding to their divergent ecological niches with respect to environmental conditions (soil moisture and other edaphic properties) [23]. Empirical studies have indicated that niche divergence and/or niche conservatism are universal in oak species at the species level, due to the impact of selection and/or adaptive differentiation when they persist in environments in which they are already well adapted [23,24,55,56], however, the niche dynamics below the species level are still elusive. Given our results indicated that these lineages have differentiated niches and their niches are not strictly conservative (Figure 5), the deep divergence time and long evolutionary history for the three intraspecific lineages suggested by Du et al. [22], and the species is separate evolving lineages (meta-population) [57,58], it therefore seems reasonable to speculate that the niches of intraspecific lineages of our focal species are undergoing divergence. Hence, if given sufficient time, the three lineages each might ultimately lead to the formation of new species.

### 4.3. Limitations and Caveats

It is noted that the plant species must migrate to track their ecological niches or increase their tolerance in situ to successfully adapt to climate change. The latter needs evidence of phenotypic plasticity and/or selection of existing phenotypes or novel phenotypes resulting from favorable mutations [59–61]. The split ENMs did not incorporate the information of plasticity or genetic adaptation, which may fail to capture the ability of each lineage to occupy other parts of the species' fundamental niche. In addition, the intraspecific lineages of *Q. aquifolioides* in the present study are assigned based on neutral genetic markers, which do not necessarily have adaptive significance. However, the genetic variation within these lineages might be influenced during long-term biogeographical processes [7,62]. Previous studies have shown that some oak species have high intraspecific genetic variation based on cpDNA markers [22,63], and greater genetic variation can allow populations or lineages to adapt to climate change through natural selection [3].

Another caveat is that ENMs are assumed to be in equilibrium with the climate [64], so that species would occupy all suitable spaces with appropriate climates after modeling. Moreover, the actual species distributions are related to dispersal limitation, interspecific interactions, soil type, land use, microscale climates, and other environmental and ecological factors [65,66]. However, given environments are variable across species distribution, and then temperature and precipitation variables can be treated as the dominant factors for projecting species ranges at regional scales [34,64], which boosts our confidence that

the results displayed here may correctly reflect the change trends of lineage distributions under contemporary climatic conditions.

## 5. Conclusions

We utilized intraspecific ENMs and niche evaluations to reveal the niche dynamics within *Q. aquifolioides* inhabiting the regions of the Sino-Himalayan Forest subkingdom. In particular, we aim to elucidate whether the three lineages (i.e., Tibet, HDM, and WSP) defined by Du et al. [22] had similar niches after their establishment via the niche similarity and niche equivalence tests implemented in the PCA-env method. Our results indicated that the three lineages had indistinguishable niches, and to some degree, their niches underwent divergence. Our findings highlight the potential to utilize split model for modeling intraspecific responses to climate change and provide insights into the diversification procedure within *Q. aquifolioides*, permitting exploration of the information determined by niche evaluations and comparisons to understand the differentiated processes of plant diversification and climate-adaptive trajectories below the species level in biodiversity hotspots.

**Supplementary Materials:** The following supporting information can be downloaded at: https://www.mdpi.com/article/10.3390/f14040690/s1, Figure S1: Phylogenetic relationships of the three intraspecific lineages in *Quercus aquifolioides* and their ancestral state reconstruction via S-DIVA analysis. Numbers above branches represent posterior probabilities, and the 95% HPD of divergence times (in Ma) of the eight main nodes (a–h) are indicated in brackets. The detailed areas related to the area delimitation in the S-DIVA analysis are shown in the left top map. Pie charts with capital letters indicate the proportions of the ancestral states. The solid lines colored in red and green with arrows indicate vicariance and long-distance dispersal, respectively (adopted from the *Journal of Biogeography* by Du et al. (2017) [22]); Figure S2: The correlated results of the 19 bioclimatic variables. Correlation coefficients are in the upper right diagonal, with their size scaled to their |r|, the bivariate scatter plots are displayed below the diagonal, while the diagonal shows histograms of the data; Figure S3: The gain values of the jackknife test using all points (real occurrences and pseudo absences) of the *Quercus aquifolioides* (QA) for training (a) and testing (b); Table S1: Sampling locations, geographical coordinates, and sample sizes for the *Quercus aquifolioides* populations included in the study of Du et al. (2017) [22]; Table S2: The variable contributions using all points (real occurrences and pseudo absences) of the *Quercus aquifolioides*; Table S3: Variable importance for the six models used in species distribution modeling. Each value is the average of 20 iterations and only variables included in the models are listed.

**Author Contributions:** Conceptualization, L.F. and X.W.; methodology, L.F.; software, L.Z.; validation, L.Z. and T.Z.; formal analysis, T.Z.; investigation, L.Z.; resources, L.F.; data curation, L.F.; writing—original draft preparation, L.F.; writing—review and editing, L.Z.; visualization, T.Z.; supervision, L.Z.; project administration, T.Z.; funding acquisition, L.F. and X.W. All authors have read and agreed to the published version of the manuscript.

**Funding:** This research was co-supported by the National Natural Science Foundation of China, grant numbers 31901075 and 31770364, and the China Postdoctoral Science Foundation, grant number 2018M633490.

**Data Availability Statement:** Not applicable.

**Acknowledgments:** The authors thank Tao Zhou, Zhonghu Li, and Guifang Zhao for their constructive feedback on an earlier draft.

**Conflicts of Interest:** The authors declare no conflict of interest.

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
