# Peer review of "Niche Dynamics Below the Species Level: Evidence from Evaluating Niche Shifts within Quercus aquifolioides"

_forests, doi:10.3390/f14040690_

Round 1
Reviewer 1 Report
The manuscript documents and investigates the environmental niche dynamic of Quercus aquifolioides in China. My comments are below:
There are certain environmental variables that distinguish the three subspecies, but these are not clear or not mentioned in your manuscript.
- You include supplementary tables for these but did not discuss it in the discussion of the manuscript. I think it would be better if these are briefly discussed or mentioned in the manuscript.
Regarding your argument on niche conservatism and how it indicates niche stability, I think niche conservatism could also be related to biotic traits? I.e., unrelated to whether these niches are stable or not. But rather, that lineages track their niche space through time and move as the environment changes. Something worth mentioning in your paper.
Minor
Fig1: Colours represent elevation but no scale is included, or also indicated in the caption either (for example low-high; green-orange/red).
Fig2: HDM or WSP, didn’t specify what these abbreviations stand for in the caption.
L92: are conservatism is grammatically incorrect, should read as ‘are conserved’.
L99: niche unfilling? I don’t get this statement, do you mean the process that is to occur or it has occurred (niche saturation).
L145-155: Sentence is too long and hard to follow. I think it would read better if you rearrange bits of the sentences to read as follows: … ‘we selected six models (xx) in the present study given the computational speed and model accuracy.’ Then start a new sentence: ‘These models look at current potential habits … etc. ‘.
And also by computational speed, I assume that you selected those models because the other models would take too long to run.
L269: confusing figure notation Fig. 5a5a56
Reviewer 2 Report
Dear authors
Your work is very interesting, but you need to work on the text (comments in the review). When reading the methodology, I had the impression that you were explaining yourself from the methodological approach. Don't do it. It seems to lack self-confidence. I hope that after your corrections the work will be accepted.
Regards
